# Asparaginase Treatment of Sea Buckthorn Berries as an Effective Tool for Acrylamide Reduction in Nutritionally Enriched Wholegrain Wheat, Rye and Triticale Biscuits

**DOI:** 10.3390/foods12173170

**Published:** 2023-08-23

**Authors:** Zuzana Ciesarová, Kristína Kukurová, Viera Jelemenská, Jana Horváthová, Janka Kubincová, Miona Belović, Aleksandra Torbica

**Affiliations:** 1National Agricultural and Food Centre, Food Research Institute, Priemyselná 4, 824 75 Bratislava, Slovakia; kristina.kukurova@nppc.sk (K.K.); viera.jelemenska@nppc.sk (V.J.); jana.horvathova@nppc.sk (J.H.); janka.kubincova@nppc.sk (J.K.); 2University of Novi Sad, Institute of Food Technology, Bulevar cara Lazara 1, 21000 Novi Sad, Serbia; miona.belovic@fins.uns.ac.rs (M.B.); aleksandra.torbica@fins.uns.ac.rs (A.T.)

**Keywords:** acrylamide, asparaginase, sea buckthorn, wheat, triticale, rye, wholegrain cereals, biscuits

## Abstract

Sea buckthorn pomace is a by-product of juice production, which is still rich in bioactive compounds. After drying, the pomace can be effectively used as a valuable addition to bakery products supporting their nutritional value. However, due to the high content of the amino acid asparagine in sea buckthorn, this promising material contributes to the undesirable formation of acrylamide. To reduce the risk from this potentially carcinogenic compound, enzymatic treatment of sea buckthorn with asparaginase was applied, which resulted in a substantial reduction of asparagine content from 1834 mg/kg in untreated dried sea buckthorn pomace to 89 mg/kg in enzymatically treated dried sea buckthorn pomace. 10% substitution of wholegrain cereal flour with enzymatically treated sea buckthorn pomace powder in rye and triticale biscuits resulted in a 35% reduction in acrylamide content, in the case of wholegrain wheat biscuits up to a 64% reduction, compared to biscuits with untreated sea buckthorn pomace powder. This study confirmed that treating fruit with asparaginase is an effective way to reduce health risk caused by acrylamide in biscuits enriched with nutritionally valuable fruit pomace.

## 1. Introduction

The current focus of consumers on healthy nutrition has brought into the spotlight many valuable sources of biologically active substances, especially of plant origin, which can be relatively easily incorporated into frequently consumed food products. Fruits or plant parts are usually used in fresh and dried form, respectively, or as extracts of individual valuable compounds [1,2,3]. The circular economy emphasizes the waste-free use of any resources, especially by-products that can be reused and, moreover, can significantly improve the nutritional profile of common food products. The implementation of selected plant-derived bioactive compounds into various food matrices is well described [4,5,6,7]. Among these valuable plants, sea buckthorn has an exceptional position due to its rich and balanced composition of polyphenols, flavonoids, carotenoids, antioxidants, vitamins, fibre, unsaturated fatty acids, essential amino acids, macro and micro elements, etc. [8,9,10,11]. Sea buckthorn pomace left over from juice production is still full of valuable bioactive substances, but due to its astringent taste, it is rarely used and consumed separately [9]. For this reason, it is a good option to use it in such matrices that provide dominant sensory attributes to food products to mask the undesirable aspects of sea buckthorn taste and aroma [12]. From this point of view, biscuits are advantageous matrices that are well accepted by consumers, are ready to eat, have a long shelf-life, with huge variabilities in composition [13]. Additionally, used cereal matrices can be valuable in their own right, if nutritionally rich wholegrain ones are used instead of commonly used wheat [14,15,16].

The necessity of heat treatment—baking—provides many positive aspects: attractive sensory profile, microbial safety and longer shelf life. These properties result from the many specific reactions within the Maillard reaction [17,18]. On the other hand, the formation of undesirable, probably carcinogenic acrylamide can be considered a disadvantage [19,20,21]. According to the current legislation [22], this process contaminant must be kept below relevant benchmark levels, which are expected to be converted to stricter maximum limits soon. This aspect is an obstacle to the wide-spread use of worthy plant by-products. Besides acrylamide, 5-hydroxymethylfurfural (HMF)—a furanic compound arising from direct dehydration of sugars under acidic conditions—is considered as potentially carcinogenic to humans [23], although no mitigation strategies specifically addressing reduction of HMF content in foods are available. Since the main mechanism of acrylamide formation is known [24,25,26,27,28], although further details are still being investigated, it seems that a good way to break this barrier is to prevent acrylamide formation by implementing some of the mitigation tools offered in Toolbox [29] and verified by many studies [30,31,32,33]. The sensory profile of final products, especially cereal-based products, is very sensitive to alterations in food processing. For this reason, the application of enzyme asparaginase is a good choice for flour treatment before baking, as it does not significantly affect the organoleptic properties of the final products. This enzymatic treatment has been successfully used mainly in cereal flour [13,34,35,36]. However, ingredients other than cereals also have the high potential to form acrylamide, in this case sea buckthorn pomace. The reason is the very high level of the main precursor—amino acid L-asparagine in sea buckthorn berries [9,37]. In order to avoid the presence of acrylamide higher than the required limit attributed to biscuits (350 µg/kg) [22], in this study, a commercial asparaginase was applied to sea buckthorn pomace before drying and subsequent baking of nutritionally enriched wholegrain cereal biscuits. This is a novel approach to prevent acrylamide formation from additional sources besides cereals. Different type of wholegrain flours from wheat, rye and triticale (hybrid of wheat and rye) were used as cereal matrices for the preparation of biscuits. The effect of enzymatically treated and untreated dried sea buckthorn pomace replacing 10% of flours was investigated for the first time to the best of our knowledge. The study followed the hypothesis that the enzymatic treatment of sea buckthorn berries leading to a reduction of asparagine content below 100 mg/kg of dried pomace powder is suitable for keeping acrylamide below the benchmark level (350 µg/kg) even in high-acrylamide rye biscuits with a 10% substitution of flour with no detrimental effect on the quality of the final biscuits.

## 2. Materials and Methods

### 2.1. Reagents and Biological Materials

#### 2.1.1. Reagents

The following enzymes, standards and chemicals were used: L-asparaginase (Novozymes, Denmark), amino acids kit of L-alanine, L-arginine hydrochloride, L-asparagine, L-aspartic acid, L-cysteine hydrochloride, L-glutamic acid, L-glutamine, L-glycine, L-histidine hydrochloride monohydrate, L-hydroxyproline, L-isoleucine, L-leucine, L-lysine hydrochloride, L-methionine, L-ornitine, L-phenylalanine, L-proline, L-serine, L-threonine, L-tryptophan, L-tyrosine and L-valine ≥ 98–99% (Sigma, St. Louis, MO, USA), acrylamide ≥ 99% (Sigma-Aldrich, Steinheim, Germany), internal standards of D3-labelled acrylamide (2,3,3-d_3_-2-propenamide ≥ 98%) and D3-labelled glutamic acid ≥ 99% (Cambridge, Isotope Laboratories, MD, USA), HPLC gradient grade solvents of acetonitrile, methanol and perfluorooctanoic acid ≥ 96% (Sigma-Aldrich, Steinheim, Germany), glacial acetic acid, ethyl acetate, potassium hexacyanoferrate trihydrate and zinc sulphate heptahydrate (Merck, Darmstadt, Germany and Lachema, Brno, Czech Republic), sodium hydrogen carbonate (Lach-ner, Neratovice, Czech Republic). Deionized water was prepared with a purification system PUR1TY Select (HP, Oxon, UK).

#### 2.1.2. Biological Material

Biscuits were produced from commercial wholegrain flours from common wheat (*Triticum aestivum*) and rye (*Secale cereale*) provided by Interpak (Kraljevo, Serbia), as well as flour obtained by milling grains of triticale (×*Triticosecale*) variety ‘Odisej’ provided by the Institute of Field and Vegetable Crops (Novi Sad, Serbia). Reconstituted wholegrain triticale flour was prepared as follows: triticale grain was milled using Quadrumat^®^ Senior roller mill (Brabender, Duisburg, Germany), and the yield of refined flour was in the range of 66–68%. Refined flour was combined with bran in the ratio 70:30 to reconstitute wholegrain triticale flour.

Further ingredients of biscuit recipe: crystal sugar (Sunoko, Novi Sad, Serbia), vanilla sugar (Dr Oetker, Bielefeld, Germany), soybean lecithin (Sojaprotein, Bečaj, Serbia) and table salt (So Produkt, Stara Pazova, Serbia) were bought at a local market. Vegetable fat HF with melting temperature of 36–38 °C (Puratos, Groot—Bijgaarden, Belgium) was used as a shortening.

Fresh sea buckthorn (*Hippophae rhamnoides* L.) berries were provided by PD Tvrdošovce farm (Tvrdošovce, Slovakia) from the crop harvested in 2020. The berries were collected on twigs, frozen at −21 °C, then plucked from the branches.

### 2.2. Methods

#### 2.2.1. Procedure of Sea Buckthorn Pomace Production

Sea buckthorn berries (5.5 kg) were washed and crushed in a hand mixer (Electro-Pulver EP8, Grâce-Hollogne, Belgium) at 13,000 rpm for 2 min. Mash (1.4 kg) was pressed and filtered manually through gauze to separate juice (1.15 kg) and pomace (0.25 kg). The sea buckthorn pomace (SBP) was dried in a drying machine (Memmert UF260, Schwabach, Germany) at 55 °C for 24 h, then ground (Grindomix GM200, Retsch, Haan, Germany) at 5000 rpm for 15 s to obtain untreated SBP powder. The remaining sea buckthorn mash (4.0 kg) underwent pH adjustment from pH 3.4 to 6.8 by adding sodium hydrogen carbonate with continuous manual stirring. pH-neutral mash (2.0 kg) was pressed and filtered as before to obtain 1.7 kg of pH-neutral juice and 0.3 kg of pH-neutral SBP which was dried using the same drying procedure as untreated SBP. Commercially produced asparaginase Acrylaway^®^L (Novozymes, Bagsværd, Denmark) provided as a gift by the producer was added to the pH-neutral wet mash (2.0 kg) at an optimized dosage of 5 mL of enzyme (3500 ASN/mL) per kg of mash. Enzyme incubation was performed in a stainless-steel container (Kitchen Aid, Artisan Series, MI, USA) for 60 min at laboratory temperature (20 °C) with low agitation. After the enzymatic treatment of the sea buckthorn mash, the procedure of pressing, filtration and drying was the same. This exact procedure of enzyme treatment of fruit and vegetable in general was submitted for industrial property protection and registered at the Office of Industrial Properties of the Slovak Republic as a utility model No 9572 [38]. Sea buckthorn juice and three samples of dried SBP powders (untreated SBP, pH neutral SBP before enzymatic treatment, and pH neutral SPB after enzymatic treatment) were collected for further characterization of amino acid profile. Two of these, untreated SBP powder and enzymatically treated pH neutral SBP powder, were used for biscuits fortification.

#### 2.2.2. Procedure of Biscuits Production

The procedure of biscuits production was developed by Belović et al. (2020) [39]. The biscuits production procedure started with mixing the fat components (shortening and soya lecithin) with crystal sugar and vanilla sugar in a mixer (Conti, model PL16 5B, Bussolengo, Italy). Then, all the powdered components were added (flour, SBP powder, sodium bicarbonate and salt). When a homogenous mixture was produced, water was added to obtain the desired degree of hydration of the dough. The dough obtained was rolled out to a thickness of 7 mm using a dough sheeter (MAC PAN, model MK 600, Thiene, Italy). The biscuits were shaped using a round mould with a diameter of 45 mm and baked in a pre-heated oven at 180 °C for 14 min (MIWE Michael Wenz GmbH, model Gusto, Arnstein, Germany). All physical measurements were performed 1 h after baking when biscuits were cooled to room temperature.

#### 2.2.3. Determination of Dimension of Biscuits

Dimensions of biscuits were measured using an analogue caliper with an accuracy of 0.01 mm for eight biscuits from each sample. Biscuit diameter (d) was divided with height (h) to calculate the spread ratio (SR) of biscuits, according to the method AACC 10–50.05 [40].

#### 2.2.4. Determination of Moisture

Moisture was determined using a rapid moisture analyzer (model HE53, Mettler Toledo, Greifensee, Switzerland). Biscuit samples were ground in a kitchen coffee grinder (Gorenje, Velenje, Slovenia) and 3 g of pulverized sample were heated at 105 °C until constant mass was reached.

#### 2.2.5. Determination of Water Activity

The water activity (Aw) was measured in two biscuits from each sample using the Aw meter Testo 205 (Testo AG, West Chester, PA, USA) according to the standard method (ISO, 2017).

#### 2.2.6. Determination of Colour

The colour of biscuits was measured using a Chroma Meter CR-400 (Konica Minolta Co., Ltd., Osaka, Japan), with D65 light source and the observer angle of 2°. Top and bottom surfaces of five biscuits per batch were measured. The results were expressed in CIE L*a*b* colour space, where CIE L* represents lightness (L* = 0—black, L* = 100—white), CIE a* represents red-green hues (+a* = redness, −a* = greenness) and CIE b* represents yellow-blue hues (+b* = yellowness, −b* = blueness).

#### 2.2.7. Determination of Texture

TA.XT Plus Texture Analyser (Stable Micro Systems, Godalming, UK) was used to determine the textural properties of biscuits. Since hardness is the most representative textural property of biscuits, it was determined as the maximum peak force recorded from the force/time curve (break force). The instrument was equipped with a 30 kg load cell and a flat blade (HDP/BS) for biscuit cutting. Instrumental settings of the standard blade cutting test for biscuits were taken from the built-in sample project BIS2_KB included in the texture analysis software Exponent v. 6.1.18.0. (Stable Micro Systems, Godalming, UK).

#### 2.2.8. Determination of Amino Acids

Determination of 22 free amino acids—alanine (Ala), arginine (Arg), asparagine (Asn), aspartic acid (Asp), cysteine (Cys), glutamine (Gln), glutamic acid (Glu), glycine (Gly), histidine (His), hydroxyproline (Hyp), isoleucine (Ile), leucine (Leu), lysine (Lys), methionine (Met), ornithine (Orn), phenylalanine (Phe), proline (Pro), serine (Ser), threonine (Thr), tryptophan (Trp), tyrosine (Tyr) and valine (Val)—was performed by a liquid chromatography-mass spectrometry (LC—MS/MS) apparatus consisting of an Agilent 1200 HPLC system (Agilent Technologies, Santa Clara, CA, USA), a binary pump, an autosampler and a temperature controlled column oven coupled to an Agilent 6410 Triple Quad detector equipped with the electrospray ionisation (ESI) interface. Samples were extracted, cleaned and directly applied directly to the LC-MS/MS analysis without prior derivatisation. The method was adapted from [41] and the exact procedure was as follows: exactly 2.0000× *g* of homogenised sample was mixed with 20 mL of acetic acid (0.2 mM) extraction solution, vortexed for 1 min and then shaken for 30 min. After centrifugation at 8720× *g* for 10 min at −5 °C the sample was diluted. 1 mL of the clear supernatant was transferred into a glass tube containing 50 µL of an internal standard solution (0.01 g of Glu-D3 in 100 mL of water) and 9 mL of acetic acid (0.2 mM) and mixed. Samples were injected after filtration through a 0.45 µm pore size nylon syringe filter (Q-Max RR Syringe Filters, Frisenette ApS, Knebel, Denmark). Analytical separation was performed on a Purospher STAR RP-8ec column (150 × 4.6 mm, 2.7 µm; Agilent, USA) using an isocratic mixture of 100 mL of acetonitrile, 500 mL of aqueous solution of PFOA (0.05 mM) and 1 mL of glacial acetic acid at a flow rate of 0.5 mL/min at ambient temperature (25 °C). The ESI mass spectrometry detection was performed in a positive mode with the following optimised parameters: drying gas temperature 320 °C; drying gas flow (N2) 8 L/min; nebulizer pressure 50 psi, capillary voltage 3 kV. Data acquisition was performed using a multiple reaction monitoring (MRM) which monitors only specific mass transitions at specific retention times (Table 1). Calibration curves of individual amino acids were prepared in the range from 0.02 to 4.00 µg/mL; an internal standard Glu-D3 concentration of 0.5 µg/mL was prepared in acidified water (0.1% acetic acid). The analysis time was 31 min. The limits of detection (LOD) were 6–10 µg/L; the limits of quantification (LOQ) were 20–30 µg/L depending on the individual amino acids (Table 1).

#### 2.2.9. Determination of Acrylamide

The samples were extracted in acetic acid (0.2 mM) water solution and pre-extracted in ethyl acetate to avoid the negative influence of salts in the chromatography system according to the previously used method [18] with a slight modification. The sample preparation was as follows: 1.0000× *g* of a homogenised sample was weighed into a capped 10 mL centrifuge tube, to which 50 µL of internal standard solution (0.0020× *g* of acrylamide-D_3_ in 100 mL of water) and 9 mL of acetic acid (0.2 mM) were added. The mixture was vortexed for 30 s and then sonicated for 5 min. Then, 500 µL of Carrez solution I (15 g of (K_4_[Fe(CN)_6_].3H_2_O in 100 mL of water) and 500 µL of Carrez solution II (30 g of (ZnSO_4_.7 H_2_O) in 100 mL of water) were added and mixed for 1 min. The mixture was then centrifuged in a Sigma 2-16KC centrifuge (Sigma, Osterode am Harz, Germany) at 8720× *g* for 10 min at −5 °C. A volume of 5 mL of the clear supernatant was transferred to a separating funnel; 5 mL of ethyl acetate was added and mixed well. The ethyl acetate layer was removed and the extraction step was repeated twice with 5 mL of ethyl acetate. 3 × 5 mL of ethyl acetate layers were collected and evaporated to dryness in a Heidolph WB 2000 vacuum rotatory evaporator (Heidolph Instruments, Schwabach, Germany) at 35 °C. The residue was dissolved in 1 mL of acetic acid solution (0.2 mM) and filtered through a 0.45 µm pore size nylon syringe filter (Q-Max RR Syringe Filters, Frisenette ApS, Knebel, Denmark). LC-MS/MS analysis was performed on a 1200 series HPLC system (Agilent Technologies, Santa Clara, CA, USA) coupled to an Agilent 6460 Triple Quad detector equipped with an ESI interface. Analytical separation was performed on an Atlantis dC18 column (100 × 3 mm, 3 µm; Waters, Milford, MA, USA) using an isocratic mixture of methanol:acetic acid:deionized water (5:1:500, *v*/*v*/*v*) at a flow rate of 0.4 mL/min at 25 °C. The ESI mass spectrometry detection was performed in positive ESI + mode with drying gas (N2) flow 8 L/min and temperature 350 °C, nebulizer pressure 50 psi, capillary voltage 2.5 kV, sheath gas flow 11 L/min at 250 °C, fragmentor 50 eV, and collision energy 10 eV. Data acquisition was performed using multiple reaction monitoring (MRM) with transition for acrylamide: 72 → 55 and acrylamide-D_3_: 75 → 58. The quantification of acrylamide was calculated from a calibration curve of the standard compound in the range from 5 to 200 ng/1 mL. The analysis time was 11 min; the retention time of acrylamide and acrylamide-D_3_ was 2.0 min. The LOD of the applied procedure was 10 µg/L; the LOQ was 15 µg/L.

#### 2.2.10. Determination of 5-Hydroxymethylfurfural

5-Hydroxymethylfurfural (HMF) was extracted with a mixture of methanol and water in a ratio of 80:20 (*v*/*v*). Approximately 1.0 g of homogeneous sample was mixed with 10 mL of the extraction mixture and sonicated for 5 min. After filtration through 0.45 µm pore size nylon syringe filters (Frisenette ApS, Knebel, Denmark) samples were ready for injection. The Agilent 1200 HPLC system (Agilent Technologies, Santa Clara, CA, USA) equipped with a quaternary gradient pump at a flow rate of 0.8 mL/min, an autosampler and a photo-diode array detector (DAD) set at 280 nm was used for the separation and quantification of HMF in the samples. The separation was performed on a Zorbax C18 SB column (250 × 4.6 mm, 5 µm; Agilent Technologies, USA). The mobile phase consisted of A (water:acetonitrile:H_3_PO_4_; 94:5:1, *v*/*v*/*v*) and B (methanol) with the following gradient elution: between 0 and 5 min: 100% A and 0% B; between 5 and 10 min: 90% A and 10% B; between 10 and 15 min: 55% A and 45% B; between 15 and 20 min: 5% A and 95% B; between 20 and 30 min: 5% A and 95% B. The samples (10.0 µL) were injected into the LC-DAD. An external calibration was used with a linearity in the range of 0.05–1.0 µg/mL. The LOD of the applied procedure was 0.5 µg/mL; the LOQ was 1.7 µg/mL. The uncertainty at the concentration level of 1.61 µg/mL was ±0.3 µg/mL.

### 2.3. Statistical Analysis

Statistica 14.0.0.15 software (Tibco Inc., Palo Alto, CA, USA, 2020, https://www.tibco.com/products/data-science (accessed on 6 January 2023) was used for statistical analysis. One-way analysis of variance (ANOVA) and Tukey’s honestly significant differences (HSD) test were used to determine the significant differences (*p* < 0.05).

## 3. Results and Discussion

### 3.1. Amino Acid Profiles of Cereal Flours

Wholegrain wheat, triticale, and rye flours were subjected to full free amino acids profile determination. Each individual amino acid was quantified, then the total amount of amino acids as well as the amount of essential and non-essential amino acids were calculated (Table 2).

Asn was identified as a dominant amino acid in all three flours (448; 782; 1183 mg/kg for wheat, triticale, and rye flour, respectively), followed by Glu (205; 218; 374 mg/kg), Arg (144; 201; 147 mg/kg), Ala (136; 117; 174 mg/kg), Asp (108; 92; 205 mg/kg), Trp (91; 40; 42 mg/kg), Gln (65; 88; 122 mg/kg), and Pro (36; 77; 163 mg/kg). For the remaining amino acids, the middle content below 50 mg/kg was observed. The highest total amount of amino acids was in rye flour (2790 mg/kg) followed by triticale flour (1939 mg/kg) and wheat flour (1520 mg/kg). The same order of flours was recognized for total amount of essential amino acids (Lys + Met + Val + Leu + Ile + Phe + Trp + Thr) and for the sum of essential and semi-essential (Arg + His) amino acid amount. This shows that wholegrain rye flour is the richest source of free amino acids, as well as essential amino acids in comparison with wholegrain wheat and wholegrain triticale flours. The content of free amino acids in triticale flour was between wheat and rye flours which is in line with the fact that triticale is a hybrid of wheat and rye. The amino acid profiles of wheat and rye flours are consistent with previously presented data [42,43,44]. Concentrations of particular amino acids in cereal varieties are affected by genetics, the environment, location, year of harvest including temperature, rainfall, pesticides, diseases and fertilization [42,45,46]. Moreover, amino acid profiles of cereal flours are dependent also on milling process (extraction rate, wet or dry heating etc.) [47]. However, the amino acid profile of triticale flour in published studies [48,49,50,51,52] did not display separately the amounts of Asp/Asn and Glu/Gln, because the analytical methods used did not allow distinguishing carboxylated and amide forms of these amino acids. In the recent studies [17,53], data for free Asn content in triticale varieties have been published, but no information on a full amino acid profile. For this reason, the complete amino acid profile data with Asp/Asn and Glu/Gln amino acids separated are unique.

Since rye typically has a high content of Asn—the main precursor of acrylamide—which is about 2.5 higher in comparison to wheat and 1.5 higher than triticale flours, it belongs to the highest donors of acrylamide in cereal-based foods. Thus, triticale flour appears to be a suitable substitute with higher nutritional value comparing to wheat and lower acrylamide formation capacity comparing to rye. Typical higher content of free Asn was observed in wholegrain flour comparing to refined flour since Asn was concentrated in outer layers of cereal grains [44]. However, the conversion of Gln and Asp to Asn catalysed by asparagine synthetase [54] can contribute to continuously elevated level of Asn in flours during processing enabled enzymatic reactions [18]. For this reason, besides Asn, the content of Gln, Glu and Asp is also important for assessment of the potential of various plant materials to form acrylamide [47]. The supporting effect of glutamine on the yield of acrylamide was described also by other authors [55]. On the other hand, cereal flours were found to contain higher concentrations of such amino acids which can reduce acrylamide level [56]. Amongst others, Pro, Trp, Cys and Gly were identified as the most effective in acrylamide inhibition [57]. Taking these aspects into account, the amounts of aforementioned amino acids were compared in all three wholegrain flours. Free Pro was substantially higher in rye and triticale flours (163.3 and 77.1 mg/kg, respectively) than in wheat one (36.1 mg/kg), contrary to Trp that was higher in wheat flour (90.6 mg/kg) than in triticale and rye (40.4 and 41.5 mg/kg, respectively) (Table 2). The amount of Gly was similar in all three flours, Cys was not detected.

### 3.2. Amino Acid Profiles of Sea Buckthorn Pomaces

Fresh sea buckthorn juice (SBJ), residual sea buckthorn pomace (SBP) after juice pressing without any treatment (SBP1), as well as SBP with neutral pH (SBP2) and enzymatically treated SBP3 were characterized by their amino acid profile (Table 2). The dominant amino acid in SBJ, SBP1 and SBP2 was Asn (1273; 1718; 1834 mg/kg) followed by Gln (109; 171; 210 mg/kg) which is approximately 10 times lower than Asn. Significant amounts were identified also for Glu (86; 124; 150 mg/kg), Asp (81; 94; 96 mg/kg), Arg (72; 142; 114 mg/kg), and Ala (62; 67; 73 mg/kg). Other amino acids in SBJ, SBP1 and SBP2 were below 50 mg/kg. The total amount of free amino acids varied between 2130 mg/kg in SBJ and 2826 mg/kg in SBP2. The cumulative amount of essential amino acids and semi-essential amino acids was between 11% (SBP2) and 17% (in SBJ) of total amount of amino acids. The highest amounts of total free amino acids (2826 mg/kg), non-essential amino acids (2508 mg/kg) and Asn (1834 mg/kg), were identified in SBP2 which is slightly higher than in SBP1. It can be supposed that free amino acids were released from protein, polypeptides and peptides as a consequence of protein hydrolysis and enzymes activity in a neutral pH environment due to application of sodium hydrogen carbonate for pH adjustment. The dominant position of Asn and a similar amino acid profile of sea buckthorn berries, juices and pomace were recognized also in other studies [11,37,58,59] contrary to the study of Zhang et al. [60] which presented Asp as the amino acid with the highest proportion. This discrepancy can be attributed to the used analytical method used in Zhang’s study which did not distinguish between carboxylated amino acids (Asp, Glu) and their amine derivatives (Asn, Gln), similarly to amino acid determination in triticale mentioned earlier. The differences in values of singular amino acid amounts can be related to different varieties, maturity stage, geographical, agronomical, harvesting and processing conditions. It is evident that due to high level of free Asn, sea buckthorn has a great capacity to form acrylamide during heat treatment. Free Asn is located in the whole berry with a higher proportion in flesh than in peel. It is supported by Asn content in SBJ, SPB1 and SBP2 (Table 2), taking into account that SBP consisting of peels and seeds after removing juice contained approximately 60% of water before drying. This means that SBP, although a highly valuable source of bioactive compounds [9,11], is also a large donor of the acrylamide precursor, even when added in small amounts. For this reason, it is fully justified to focus on reducing the content of asparagine in all raw materials used in food processing, not only in cereals. The effectiveness of enzymatic treatment performed on homogenized sea buckthorn mash after pH-adjustment was evident in the Asn and Asp values. After 60 min of enzyme treatment, the striking changes in the mentioned amino acid proportions were observed. The Asn content dropped from 1834 mg/kg to 89 mg/kg with a parallel increase of Asp from 96 mg/kg to 1470 mg/kg. No other significant changes in particular amino acid content before and after enzymatic treatment were noticed (Table 2), neither in Gln and Glu amounts, which pointed to the excellent specificity of the asparaginase. Since Gln has a similar structure to Asn, some asparaginases also have a low activity towards Gln [61]. Because the glutaminase activity can have a serious detrimental effect on human health (liver dysfunction, pancreatitis, leucopoenia) [62], this specific type of asparaginase with known glutaminase activity should not be used for food treatment.

### 3.3. Characterization of Cereal Biscuits Enriched with SBP Powder

Cereal biscuits are often used as a favourable model system for several reasons. In experiments, they are good matrices for implementations of different ingredients with simple reproducibility of well-defined processing and long shelf-life. As products, they are frequently consumed in countless variations and well-accepted by consumers [13]. In this experiment, cereal biscuits from wholegrain wheat, triticale and rye flours were prepared according to the procedure described in 2.1.4 (as controls), and two kinds of SBP powders—untreated SBP1 and enzymatically treated pH-neutral SBP3—were incorporated into the recipe as a 10% substitution of the respective flour. The impact of SBP presence in cereal biscuits was examined by characterisation of spread ratio, moisture, water activity, colour, texture, acrylamide content, residual free amino acid content and HMF content.

#### 3.3.1. Impact of SBP Powders on Dimensions of Biscuits

Although all biscuits were prepared by the same procedure with the same diameters and heights, the differences in dimension parameters (expressed as a spread ratio value) in final biscuits were observed (Table 3). The higher spread ratio of rye and triticale biscuits in comparison to wheat biscuits can be the consequence of their specific proteins [63]. Doughs prepared from these flours have lower viscosity, resulting in a higher spread ratio. The increase in spread ratio observed in biscuits prepared with SBP1 and SBP3 can be related to the dilution of gluten in biscuit doughs by dietary fibre and subsequent lower viscosity, which was also observed in a paper by Tomić et al. (2016) [64]. Namely, dried SBP was previously shown to be a raw material rich in crude fibre (19.86%) [65]. However, there were no substantial differences between the spread ratios of biscuits prepared with untreated SBP1 and enzymatically treated SBP3. This is in accordance with the results of Gazi et al. (2023) [13], who also observed that the addition of asparaginase in wire-cut cookie dough prepared from wheat flour did not cause a significant difference in the spread ratio. This result is important for the industrial production of biscuits, which can be automated only if the dimensions of biscuits have low variability [66].

#### 3.3.2. Impact of SBP Powders on Moisture and Water Activity of Biscuits

Values of moisture varied between 0.9 and 6.6% and water activity (Aw) was in the range of 0.29 to 0.54 (Table 3). Lower moisture content and lower Aw values in biscuits prepared with SBP1 and SBP3 can be related to the lower height of these biscuits. Biscuits with lower heights have consequently higher spread ratio and thus larger surfaces exposed to the heat in the oven, making the evaporation process during baking faster within the same baking time.

#### 3.3.3. Impact of SBP Powders on the Colour of Biscuits

The colour of biscuits is influenced both by the colour of the ingredients used and the baking process. The main colour characteristics from both upper and lower surfaces of final biscuits expressed as a* (redness), b* (yellowness) and L* (lightness) are summarized in Table 3.

It is known that the colour of wholegrain flours is more or less determined by the presence of anthocyanin pigments [67], but flavonoids as a subgroup of anthocyanin pigments are present in sea buckthorn in higher level than in flours (especially from non-coloured grains). Anthocyanins can be found in different chemical forms which depend on the pH of the solution. At pH values higher than 7, the anthocyanins are degraded depending on their substituent groups [68]. Therefore, the increase in red tone (a*) observed after substitution by SBP1 can be related to pH lower than 7, whereas the lower a* values in enzymatically treated SBP3 biscuits compared to SBP1 biscuits can be related to pH values close to 7, which are required for proper asparaginase activity, and subsequent degradation of anthocyanins.

Additionally, brown polymers called melanoidins are formed as Maillard reaction products during baking [17]. Therefore, an increase in red tone (a*) observed after substitution by SBP1 can be also related to the more intensive Maillard reactions as a consequence of the high content of amino acids, especially Asn, in SBP1. Consequently, lower a* values in enzymatically treated SBP3 biscuits compared to SBP1 biscuits can be partly attributed to the action of asparaginase, which reduce the total amount of free amino acids. The substantial increase of yellow tone (b*) in both SBP1 and SBP3 biscuits can be related to the presence of β-carotene and zeaxanthin, the main pigments found in dried SBP [65].

#### 3.3.4. Impact of SBP Powders on Texture of Biscuits

Generally, the texture characteristics of biscuits are strongly dependent on ingredients and processing conditions thus changes in texture are legitimately expected. The hardness of biscuits is influenced by multiple factors, including protein properties, amount of dietary fibre and particle size of flour used for their preparation, as well as by processing conditions. In our previous study [39], similarly to the presented study (Table 3), cookies prepared from wholegrain triticale flour were shown to be significantly harder than wholegrain wheat and rye cookies, supposedly due to the high content of coarse bran particles. These coarse bran particles were obtained during the milling process by roller mill, which differs from the process used in the production of commercial wholegrain wheat and rye flours. Based on our observations, the substitution of wholegrain flours by SBP1 powder did not significantly influence the hardness of the biscuits. However, the substitution by enzymatically treated SBP3 powder led to a significant decrease in hardness. Although it seems enzymatic treatment had a visible effect on textural properties of biscuits, there is also another aspect which could be taken into consideration. Since enzymatically treated SBP3 had its pH value adjusted to 6.8 to make the pH value closer to asparaginase optimum, contrary to pH value 3.4 measured in untreated SBP1, it can be concluded that this change in pH value can influence the textural properties of biscuits. The significant decrease in the hardness of biscuits with higher concentrations of sodium hydrogen carbonate and potassium hydrogen carbonate in the dough was observed by Chen et al. (2020) [69]. They concluded that the inclusion of high levels of KHCO_3_ stimulated gluten aggregation via non-redox reactions, i.e., not via SH oxidation. Based on their results, it could be assumed that in a more alkaline environment, more proteins are aggregated via non-covalent bonds, which are weaker and lead to a less stable and less compact protein network. Therefore, they are more easily disaggregated during processing and additional binding of water within the system occurs, resulting in the considerable softening of biscuits. During baking, the water in the protein matrix remains bound longer than it is bound to the dietary fibre, so a biscuit with fewer S-S bonds is softer than one with more S-S bonds.

#### 3.3.5. Impact of SBP Powders on Acrylamide and Residual Free Asparagine in Biscuits

Acrylamide was formed in biscuits prepared from wheat, triticale and rye flours without and with the addition of SBP powders in different amounts. In control biscuits without SBP powder addition (SBP0), acrylamide was detected in levels related to the predisposition of flours used for biscuits preparation. The lowest acrylamide level (27.2 µg/kg DW) was detected in wheat biscuits, followed by triticale biscuits (58.9 µg/kg DW), and the highest acrylamide level was observed in rye biscuits (105.3 µg/kg DW) (Table 4). It is in a good agreement with Asn level in relevant flours which were 448; 782; 1183 mg/kg for wheat, triticale and rye flours, respectively, with a linear correlation between free Asn in dough and acrylamide in biscuits content (R^2^ = 0.9970) (Figure 1a).

A 10% substitution of flours with SBP1 powder resulted in a substantial increase of acrylamide level in all biscuits (80.4; 106.0; 485.5 µg/kg DW for wheat, triticale and rye biscuits) which was from 2 to 5 times higher than in the respective control biscuits SBP0 (Table 4).

The increased level of acrylamide in biscuits with SBP1 powder is expected since the higher amount of the main precursor—free Asn—entered the Maillard reaction responsible for acrylamide formation. Untreated SBP1 powder is very high in Asn, thus it is a significant donor of Asn in dough. The amount of available Asn in biscuit dough enriched with SBP1 addition increased by 7% in rye biscuits, by 17% in triticale biscuits and by 30% in wheat biscuits, however, the increase of acrylamide in respected biscuits was substantially higher with exponential correlation between free Asn content in SBP1 biscuit dough and acrylamide (R^2^ = 0.9666) (Figure 1a). For this reason, the observations of Borczak et al. [6] declaring the 89% drop of acrylamide after 5% sea buckthorn application in wheat flour cookies, as well as other wild-grown fruits (hawthorns, rowans, wild roses, elderberries, chokeberries), is surprising since many of them are also high in Asn content. Authors attributed this phenomenon to the bioactive molecules present in the tested wild fruits. Actually, in their recipe they replaced only 1% of flour with a lyophilized fruit powder, although they mistakenly declared 5% substitution, which represented only 0.5% in a total dough amount. The effect of additives with antioxidative properties on acrylamide formation is usually dose-dependent [70] and supporting or suppressive effects are ambiguous. A significantly higher tendency of sea buckthorn puree to form acrylamide under thermal conditions was unequivocally demonstrated in a model system [37]. In other studies [70,71], a slight increase of acrylamide level between 10 to 20% was observed in the crumb of bakery products with 5% sea buckthorn powder addition, but 2–5 times higher acrylamide was observed at 10% sea buckthorn powder addition. However, in these studies, no information on Asn content in matrices was presented. Moreover, in these studies, acrylamide was determined by other analytical techniques (HPLC method with a UV-Vis detector [6], capillary electrophoresis system [71,72]) instead of a highly selective and sensitive LC-MS/MS technique which is predominantly used for acrylamide determination [73,74]. On the other hand, the HPLC-UV technique used for acrylamide determination with acceptable parameters was demonstrated in the case of acrylamide detection in olives [75] and deep-fried flour-based foods [76].

Enzymatic treatment of SBP with asparaginase after previous pH adjustment according to the procedure described in the patent application [38] used in this study, resulted in a substantial decrease of free Asn in SBP from 1834 mg/kg to 89 mg/kg (Table 2) which was lower than the free Asn content in used wholegrain flours. The 10% substitution of flours by SBP3 meant a decrease of free Asn in doughs available for acrylamide formation. Thus, acrylamide level in biscuits with enzymatically treated SBP3 powder was lower (29.1; 69.1; 316.6 µg/kg for wheat, triticale and rye biscuits) comparing to acrylamide in SBP1 biscuits (Table 4). However, despite expectations of similar acrylamide content in SBP3 and SBP0 biscuits, the decrease of free Asn in triticale and rye doughs did not project directly, so acrylamide decreased in the respective biscuits. The correlation of Asn in dough and acrylamide level in biscuits was not linear (e.g., in the case of no SBP addition, but a good exponential correlation was achieved (R^2^ = 0.9966), similar to Asn/acrylamide correlation in case of SBP1 dough and biscuits (Figure 1a). It can be supposed that not only free Asn level is responsible for acrylamide formation in complex matrices, but also other factors affected the effectiveness of acrylamide formation. This is evident from a weak linear correlation of free Asn in doughs and final acrylamide content of SBP0, SBP1 and SBP3 biscuits (R^2^ = 0.5684) (Figure 1b). This unproportional content of acrylamide can be probably caused by a richer matrix represented by SBP containing antioxidants, flavonoids, carotenes and a high portion of lipids. Some of these substances can support acrylamide formation through other pathways than a Maillard reaction, but also, they can suppress it. Moreover, the opposite reactions may result from the presence of similar factors in flour which inhibit acrylamide formation. Amongst others, the significantly higher concentration of free amino acids Pro, Trp and Gly in flours compared to SBPs are notable which may inhibit acrylamide formation or react with acrylamide and reduce its levels [42,57]. These opposite reactions resulted in less acrylamide formed per unit of Asn at higher presence of these amino acids, notably visible in triticale and rye SPB0 biscuits comparing to SPB3 biscuits (Table 4). The different pH of dough before baking due to the presence of untreated SBP1 powder with pH 3.4 and enzymatically treated SBP3 powder with pH 6.8 can also play the role. This hypothesis with multifactorial impact of matrix on acrylamide formation is supported by determination of residual free Asn in biscuits after baking.

In control biscuits without addition of SBP powder, the highest level of residual free Asn after baking was observed in rye biscuits (342 mg/kg DW), followed by triticale (193 mg/kg DW) and wheat biscuits (157 mg/kg DW) (Table 4), which was in the same order as free Asn in particular flours before baking. It means that only a partial Asn utilisation was observed and the residual Asn represent 40–56% of the initial free Asn in dough. This was evident by comparison of Asn content calculated from relevant flours and determination of free Asn in biscuits after baking. The similar utilization of Asn was observed in case of the 10% substitution of wholegrain flours by untreated SBP1 powder as shown by determination of residual Asn in SPB1 biscuits. The residual Asn in these biscuits represented 41–50% of the initial Asn content. In biscuits with enzymatically treated SBP3 powders, the utilization of free Asn was higher, since the residual Asn was between 17% and 28% of the initial Asn content. As it was mentioned earlier, the asparaginase treatment was accompanied with pH adjustment to neutral values, thus under these conditions the conversion of Asn to acrylamide was more effective. It is known that the pH value plays an important role in acrylamide formation [77] which decreased in a buffered system [78].

#### 3.3.6. Impact of SBP Powders on HMF of Biscuits

Formation of HMF is supported by lower pH and the presence of acidic matrix [33,78]. In case of untreated SBP1 powder, the pH of SBP before drying is only 3.4 due to a high content of organic and phenolic acids [9,65]. Addition of untreated SBP1 powder to dough of cereal biscuit production resulted in higher HMF content in final biscuits. It can be assumed that HMF formation during baking was promoted by lower pH due to the acidic nature of SBP1. This impact of pH was supported with the observation of HMF detected only in biscuits with untreated SBP1 powder addition (8.13; 5.25 and 6.58 mg/kg DW for wheat, triticale and rye biscuits) (Table 4). In control biscuits without SBP powder addition (SBP0) and in biscuits with enzymatically treated SBP powder addition (SBP3), the HMF was not detected.

## 4. Conclusions

Following the trend to reuse valuable by-products of plant-based food processing, sea buckthorn rich in exceptionally high levels of health-promoting compounds is worthy of use in commonly well accepted cereal biscuit production. The disadvantage of its high potential to contribute to undesirable acrylamide formation can be eliminated by enzymatic treatment of SBP before its usage in biscuit production. Asparaginase application on SBP under pH neutral conditions decreased free Asn up to 95% from 1834 mg/kg to the final Asn concentration of 89 mg/kg. The 10% substitution of wholegrain cereal flours (wheat, triticale, rye) by enzymatically treated SBP resulted in a substantial reduction of acrylamide in enriched biscuits between 30 and 60% without detrimental effects on quality of final products. Even in high-acrylamide rye biscuits with SBP addition, the enzymatic treatment of SBP brought a decrease of acrylamide from 485.5 µg/kg to 316.6 µg/kg which met the benchmark level established for biscuits and wafers (350 µg/kg). This study confirmed that the enzymatic treatment is a proper tool for acrylamide mitigation not only in cereal based processed foods, but also in fruits and fruit wastes used as nutritionally valuable additives of enriched cereal products.

## Figures and Tables

**Figure 1 foods-12-03170-f001:**
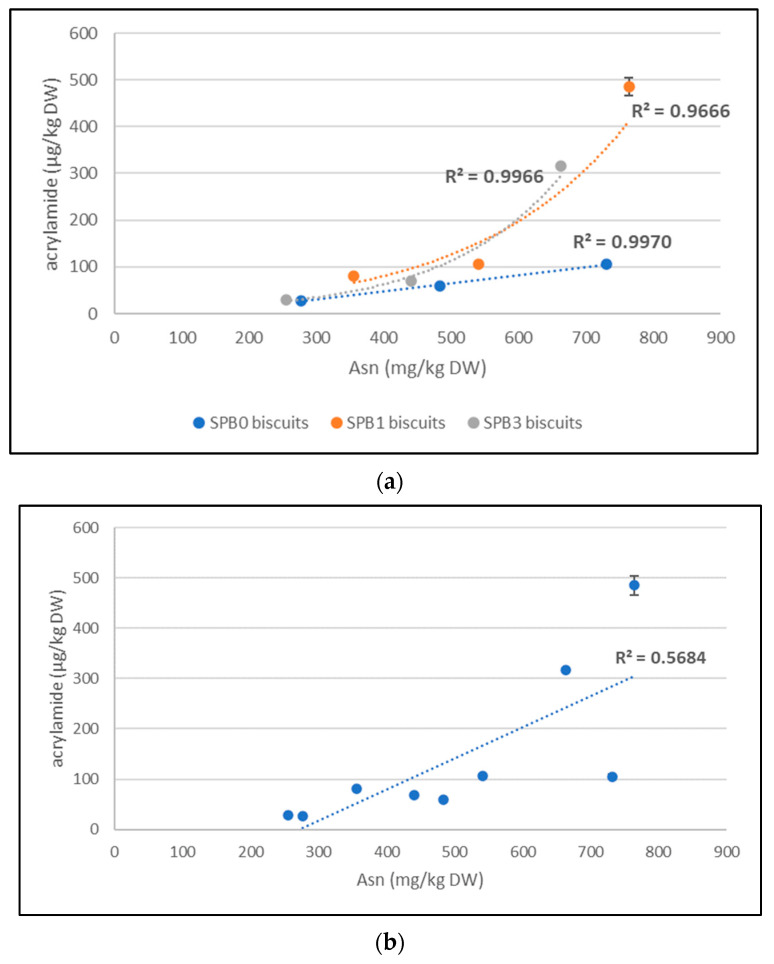
(**a**,**b**). Correlation of free asparagine (mg/kg DW) in dough and acrylamide (µg/kg DW) in respected biscuits. (**a**)—separate correlations of cereal biscuits SBP0, SBP1 and SBP3; (**b**)—correlation not distinguishing between cereal biscuits and SBP addition.

**Table 1 foods-12-03170-t001:** Optimized MRM parameters and parameters of calibration for LC/MS-MS determination of amino acids.RT—Retention time; PI—Precursor Ion; Q1—Qualifier; Q2—Quantifier; R^2^—Coefficient of Determination; LOD—Limit of Detection; LOQ—Limit of Quantification.

Amino Acid	RT (min)	PI	Production Ions		Fragmentor (eV)	Collision Energy (eV)		Dwell (ms)	R^2^	LOD (ng/mL)	LOQ (ng/mL)
			Q1	Q2		Q1	Q2				
Ala	4.53	90	44	90	50	6	2	20	0.9990	10	30
Arg	13.64	175	70	175	100	25	5	50	0.9992	10	30
Asn	4.03	133	74	87	50	12	5	20	0.9998	8	25
Asp	3.74	134	74	88	50	12	5	20	0.9996	10	30
Cys	3.92	122	76	59	60	10	25	20	0.9991	10	30
Gln	4.09	147	84	130	50	15	5	20	0.9919	10	30
Glu	4.21	148	84	102	50	15	10	20	0.9994	6	20
Gly	4.47	76	76	30	50	2	6	20	0.9997	11	33
His	11.67	156	110	156	100	15	2	50	0.9996	10	29
Hyp	3.63	132	86	132	50	15	2	20	0.9998	13	39
Ile	7.62	132	86	132	80	5	2	50	0.9997	8	24
Leu	7.11	132	86	132	80	5	2	50	0.9997	11	33
Lys	13.28	147	84	130	80	15	5	50	0.9991	7	23
Met	5.45	150	56	133	80	15	5	50	0.9972	4	12
Orn	12.35	133	116	70	80	5	20	50	0.9997	9	28
Phe	8.21	166	120	103	80	10	30	50	0.9997	10	30
Pro	3.95	116	70	116	50	10	0	5	0.9978	9	29
Ser	4.1	106	60	106	50	5	2	20	0.9987	10	30
Thr	4.18	120	56	74	50	15	8	20	0.9978	8	26
Trp	15.17	205	146	188	100	15	5	50	0.9998	11	34
Tyr	8.14	182	182	136	100	2	10	50	0.9979	10	30
Val	5.32	118	72	118	80	10	2	50	0.9994	3	10

**Table 2 foods-12-03170-t002:** Amino acid content (mg/kg) in wholegrain cereal flours, sea buckthorn juice and pomaces.

		Wholegrain Flour			Sea Buckthorn Pomace		
Amino Acid (mg/kg)	Wheat	Triticale	Rye	SBJ	SBP1	SBP2	SBP3
Asn	447.9 ^c^	782.0 ^b^	1182.9 ^a^	1273.3 ^C^	1717.8 ^B^	1833.9 ^A^	88.5 ^D^
Asp	107.9 ^b^	92.2 ^c^	205.1 ^a^	81.2 ^B^	94.1 ^B^	95.6 ^B^	1469.5 ^A^
Gln	65.2 ^c^	87.6 ^b^	121.6 ^a^	109.0 ^C^	170.8 ^B^	209.7 ^A^	215.5 ^A^
Glu	205.2 ^c^	218.4 ^b^	373.9 ^a^	86.2 ^C^	124.2 ^B^	150.6 ^A^	127.7 ^B^
Arg	143.6 ^b^	200.6 ^a^	146.8 ^b^	72.2 ^B^	141.9 ^A^	113.9 ^A^	110.3 ^A^
Lys	49.5 ^c^	63.7 ^a^	58.7 ^b^	38.8 ^A^	29.6 ^B^	30.5 ^B^	26.8 ^B^
Ala	135.7 ^b^	116.8 ^c^	173.7 ^a^	62.3 ^B^	67.4 ^AB^	72.6 ^A^	63.7 ^B^
Phe	26.4 ^b^	42.2 ^a^	41.6 ^a^	86.6 ^A^	42.7 ^B^	35.6 ^B^	32.3 ^B^
Pro	36.1 ^c^	77.1 ^b^	163.3 ^a^	39.9 ^B^	48.9 ^A^	48.9 ^A^	42.5 ^B^
Trp	90.6 ^a^	40.4 ^b^	41.5 ^b^	22.8 ^B^	27.2 ^A^	23.6 ^B^	23.0 ^B^
Ser	28.9 ^a^	18.5 ^b^	21.3 ^b^	39.8 ^A^	53.3 ^A^	47.2 ^A^	52.0 ^A^
Val	36.6 ^c^	46.4 ^b^	82.1 ^a^	21.4 ^A^	22.2 ^A^	22.8 ^A^	22.4 ^A^
Met	0.9 ^c^	1.6 ^b^	7.3 ^a^	13.3 ^A^	11.6 ^BC^	11.9 ^B^	11.2 ^C^
Tyr	nd	nd	nd	41.8 ^A^	26.0 ^B^	27.6 ^B^	24.8 ^B^
Ile	40.0 ^b^	48.3 ^a^	47.7 ^a^	38.5 ^A^	17.6 ^B^	19.8 ^B^	19.5 ^B^
Leu	13.9 ^c^	16.4 ^b^	25.9 ^a^	9.5 ^B^	11.6 ^AB^	11.8 ^AB^	12.6 ^A^
Thr	18.3 ^c^	20.9 ^b^	25.0 ^a^	22.7 ^A^	21.3 ^A^	23.4 ^A^	21.3 ^A^
Gly	53.1 ^a^	42.1 ^b^	44.1 ^b^	11.3 ^A^	14.3 ^A^	11.8 ^A^	12.0 ^A^
His	17.1 ^c^	19.5 ^b^	24.1 ^a^	31.7 ^A^	30.9 ^A^	24.6 ^B^	25.1 ^B^
Orn	3.1 ^b^	4.6 ^a^	3.5 ^b^	10.3 ^AB^	14.3 ^A^	8.8 ^B^	nd
Cys	nd	nd	nd	16.3 ^A^	16.7 ^A^	nd	15.5 ^A^
Hyp	nd	nd	nd	1.1 ^A^	1.2 ^A^	1.3 ^A^	1.0 ^A^
Total	1519.9	1939.3	2790.4	2130.1	2705.7	2826.2	2417.3
E-AA	276.0	280.0	330.0	253.6	183.7	179.6	169.1
SemiE-AA	160.7	220.1	170.9	103.9	172.9	138.5	135.4
NonE-AA	1083.0	1439.2	2289.5	1772.5	2349.2	2508.1	2112.2

Values are the average (*n* = 4); nd = not detected; relative standard deviations are below 5% for Asn, Asp, Gln, Glu, Lys, Ala, Pro, Trp, Ser, Val, Ile, Leu, Thr, Gly, His, Cys, Hyp; and below 10% for Arg, Phe, Met, Orn. Different letters (a–c) indicate significant differences (*p* < 0.05) among different cereal flours, and (A–D) suggest significant differences (*p* < 0.05) between different sea buckthorn products. SBJ—fresh sea buckthorn juice; SBP1—residual sea buckthorn pomace after juice pressing without any treatment; SBP2—residual sea buckthorn pomace after juice pressing with neutral pH; SBP3—residual sea buckthorn pomace after juice pressing enzymatically treated at neutral pH; E-AA—essential amino acids; SemiE-AA—semi essential amino acids; NonE-AA—nonessential amino acids.

**Table 3 foods-12-03170-t003:** Characteristics of wholegrain cereal biscuits with sea buckthorn pomace powder addition.

Wholegrain Cereal Biscuits with SBP	Wheat			Triticale			Rye		
	SBP0	SBP1	SBP3	SBP0	SBP1	SBP3	SBP0	SBP1	SBP3
Spread ratio(−)	4.70 ^bcA^	4.32 ^cdB^	4.45 ^bcdAB^	4.05 ^dC^	4.68 ^bcB^	5.59 ^aA^	4.95 ^bB^	5.54 ^aA^	5.85 ^aA^
Weight (g)	8.51 ^cA^	7.23 ^eB^	7.39 ^eB^	8.48 ^cB^	9.45 ^aA^	6.70 ^fC^	8.85 ^bA^	8.20 ^dB^	6.78 ^fC^
Moisture (%)	6.20 ^aA^	5.09 ^bB^	3.84 ^cC^	6.59 ^aA^	2.78 ^dC^	4.66 ^bB^	3.12 ^dA^	1.22 ^eB^	0.93 ^eC^
Aw value	0.475 ^cA^	0.189 ^iC^	0.382 ^fB^	0.539 ^aA^	0.342 ^gC^	0.441 ^dB^	0.513 ^bA^	0.420 ^eB^	0.290 ^hC^
Colour									
L* (upper)	60.60 ^aA^	54.95 ^bB^	55.31 ^bB^	52.84 ^bcA^	50.73 ^cdeAB^	49.33 ^defB^	52.42 ^bcdA^	47.36 ^efB^	46.01 ^fB^
a* (upper)	5.99 ^dB^	9.40 ^bcA^	8.19 ^cA^	8.93 ^bcB^	10.53 ^abB^	9.35 ^cA^	7.96 ^cB^	11.35 ^aA^	10.14 ^abA^
b* (upper)	24.93 ^dB^	36.33 ^aA^	37.49 ^aA^	24.22 ^dC^	33.38 ^bA^	29.91 ^cB^	25.91 ^dB^	31.60 ^bcA^	30.37 ^cA^
L* (lower)	58.83 ^aA^	52.72 ^bB^	51.32 ^bcdB^	52.51 ^bA^	46.64 ^eB^	48.29 ^cdeB^	52.29 ^bcA^	45.08 ^eC^	47.33 ^deB^
a* (lower)	6.95 ^fB^	10.87 ^cdA^	8.50 ^efB^	9.16 ^deB^	13.16 ^abA^	10.48 ^cdeB^	9.65 ^cdeC^	13.42 ^aA^	11.36 ^bcB^
b* (lower)	26.53 ^dB^	37.10 ^aA^	35.19 ^abA^	25.38 ^dB^	32.46 ^bcA^	31.21 ^cA^	28.00 ^dB^	33.35 ^bcA^	33.30 ^bcA^
Texture									
Hardness (kg)	7.80 ^bcA^	9.40 ^bA^	7.26 ^bcA^	16.98 ^aA^	17.18 ^aA^	10.45 ^bB^	10.60 ^bA^	8.41 ^bcA^	4.13 ^cB^

Values are the average: spread ratio and weight (*n* = 8); moisture and Aw (*n* = 2); colour and hardness (*n* = 5). Relative standard deviations are below 5% for spread ratio, weight, moisture, Aw, L*, b*; below 10% for a*; below 20% for hardness. Different letters (a–h) indicate significant differences (*p* < 0.05) in a row, and (A–C) suggest significant differences (*p* < 0.05) within the group of cereal biscuits with different sea buckthorn pomaces. SBP0—biscuits with no presence of sea buckthorn pomace powder; SBP1—biscuits with sea buckthorn pomace powder without any treatment; SBP3 biscuits with sea buckthorn pomace powder enzymatically treated at neutral pH.

**Table 4 foods-12-03170-t004:** Acrylamide (µg/kg DW), initial Asn (mg/kg DW), residual Asn (mg/kg DW) and HMF (mg/kg DW) of cereal biscuits with sea buckthorn pomace powder addition.

Cereal Biscuits with SBP	Wheat			Triticale			Rye		
	SBP0	SBP1	SBP3	SBP0	SBP1	SBP3	SBP0	SBP1	SBP3
Acrylamide (µg/kg DW)	27.2 ^eB^	80.4 ^dA^	29.1 ^eB^	58.9 ^dC^	106.0 ^cA^	69.1 ^dB^	105.3 ^cC^	485.5 ^aA^	316.6 ^bB^
Calculated initial Asn (mg/kg DW)	276.9	355.4	254.7	483.5	541.3	440.6	731.3	764.3	663.7
Residual Asn (mg/kg DW)	156.6 ^cdB^	189.3 ^bcdA^	72.9 ^dC^	193.4 ^bcdAB^	256.8 ^abcA^	76.7 ^dB^	342.0 ^aA^	327.1 ^abA^	164.5 ^cdA^
Acrylamide per unit of Asn	0.10	0.23	0.11	0.12	0.20	0.17	0.14	0.64	0.48
HMF (mg/kg DW)	nd	7.72 ^a^	nd	nd	5.25 ^c^	nd	nd	6.58 ^b^	nd

Values are the mean: acrylamide (*n* = 3); Asn, HMF (*n* = 2). Relative standard deviations are below 5% for acryalmide and HMF, below 10% for Asn. Different letters (a–e) indicate significant differences (*p* < 0.05) in a row, and (A–C) suggest significant differences (*p* < 0.05) within the group of cereal biscuits with different sea buckthorn pomaces. SBP0—biscuits with no presence of sea buckthorn pomace powder; SBP1—biscuits with sea buckthorn pomace powder without any treatment; SBP3 biscuits with sea buckthorn pomace powder enzymatically treated at neutral pH.

## Data Availability

The data used to support the findings of this study can be made available by the corresponding author upon request.

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
