# Peer review of "Asparaginase Treatment of Sea Buckthorn Berries as an Effective Tool for Acrylamide Reduction in Nutritionally Enriched Wholegrain Wheat, Rye and Triticale Biscuits"

_foods, 2023, doi:10.3390/foods12173170_

Round 1
Reviewer 1 Report
Comments and Suggestions for Authors
The manuscript describes the influence of asparaginase treatment on the acrylamide content of biscuits fortified with sea buckthorn berries pomace. The topic is interesting and promising results are reported. However, some revisions are required:
Comments:
- Line 155: Write the model and accuracy of the caliper.
-Line 165: water activity ( Correct it throughout the entire manuscript)
- Line 178: Which textural test was performed? Was it triple point bend?
- Please add more discussion about the effect of cereal type on different properties of biscuits (e.g. color and texture)
Comments on the Quality of English Language
The language is fine
Reviewer 2 Report
Comments and Suggestions for Authors
The manuscript follows the asparaginase treatment of a by-product of sea buckthorn berries as a tool to reduce the concentration of acrylamide in nutritionally enriched biscuits. The subject is very interesting but there are some concerns regarding some aspects from the manuscript that have to be taken into account.
Introduction - ''trendy'' - is a colloquial word so please replace it with an appropriate word. A phrase should be added to state the novelty of the study for the targeted field of research and also why were biscuits chosen as the matrix to incorporate the pomace? Again, please eliminate the first person point of view from your manuscript and change it to a third person. instead we did, the study followed etc..
Materials and methods
2.1.2. Biological Material - the species of the plant should be written in italic. Please specify the origin of the commercial flours. Line 124 - underwent instead of was undergone.
2.2.1. Procedure of Sea Buckthorn Pomace Production - "Commercially produced asparaginase Acrylaway®L (Novozymes, Den-128 mark) complimentary provided by the producer was applied into the pH-neutral wet 129 mash (2.0 kg) in a dosage of 5 mL of enzyme (3500 ASN/mL) per kg of mash" - it is not clear the basis on which the authors have chosen these values for the enzyme. Please explain your choice and would be really interesting to correlate with the activity of the enzyme. Has the activity of the enzyme been determined?
2.2.2. Procedure of Biscuits Production - Is the recipe adapted or its a personal recipe?
2.2.8. Determination of Amino Acids - Line 197 - 2.000g? what type of value is this? Unclear if it's 2g. Please rectify this.
3. Results and Discussion
3.1. Amino Acid Profiles of Cereal Flours - the authors state that they employed an LC/MS-MS method to identify the amino acids from the samples but there is no data regarding the m/z of each AA or the retention times nor the peak area. I think the table should be completed with these factors and parameters of the chromatographic analysis. Furthermore, it would have been very interesting to see the concentration of each amino acid of the dough of each type of biscuits before the baking. Can the authors provide any information on the matter?
Comments on the Quality of English Language
The English level of the manuscript is good but just some corrections must be made regarding the English grammar:
Use the forms "a", "an" or "the" in order to define the word. Remember, to use A, AN, and THE properly, you must know whether or not a noun is a Count or Non-Count Noun. (A count noun is the name of something that can be counted: one experiment, two experiments, three experiments. A non-count noun is the name of something that cannot be counted: milk, flour, freedom, justice, food).
Check each verb to see if it has the right form and check the tenses you want to use. Pay attention which tense you want to use if you have present tense for example, stick to this tense throughout the whole sentence.
Use the correct form of the verb as requested by the used noun (Pay attention if the noun is countable or not).
Round 2
Reviewer 2 Report
Comments and Suggestions for Authors
The authors have complied and answered all the comments so I recommend the acceptance of the manuscript in its present form.